# Mining Indonesian Microbial Biodiversity for Novel Natural Compounds by a Combined Genome Mining and Molecular Networking Approach

**DOI:** 10.3390/md19060316

**Published:** 2021-05-28

**Authors:** Ira Handayani, Hamada Saad, Shanti Ratnakomala, Puspita Lisdiyanti, Wien Kusharyoto, Janina Krause, Andreas Kulik, Wolfgang Wohlleben, Saefuddin Aziz, Harald Gross, Athina Gavriilidou, Nadine Ziemert, Yvonne Mast

**Affiliations:** 1Department of Microbiology/Biotechnology, Interfaculty Institute of Microbiology and Infection Medicine, Tübingen (IMIT), Cluster of Excellence ‘Controlling Microbes to Fight Infections’, University of Tübingen, Auf der Morgenstelle 28, 72076 Tübingen, Germany; irahndyn@gmail.com (I.H.); Janina.Krause-d1j@rub.de (J.K.); andreas.kulik@uni-tuebingen.de (A.K.); wolfgang.wohlleben@biotech.uni-tuebingen.de (W.W.); 2Research Center for Biotechnology, Indonesian Institute of Sciences (LIPI), Jl. Raya Jakarta-Bogor KM.46, Cibinong, West Java 16911, Indonesia; puspita.lisdiyanti@bioteknologi.lipi.go.id (P.L.); wien.kyoto@gmail.com (W.K.); 3Department of Pharmaceutical Biology, Institute of Pharmaceutical Sciences, University of Tübingen, Auf der Morgenstelle 8, 72076 Tübingen, Germany; Hamada.saad@pharm.uni-tuebingen.de (H.S.); azizgene@gmail.com (S.A.); harald.gross@uni-tuebingen.de (H.G.); 4Department of Phytochemistry and Plant Systematics, Division of Pharmaceutical Industries, National Research Centre, Dokki, Cairo 12622, Egypt; 5Research Center for Biology, Indonesian Institute of Sciences (LIPI), Jl. Raya Jakarta-Bogor KM.46, Cibinong, West Java 16911, Indonesia; shanti_ratna01@yahoo.com; 6Applied Natural Products Genome Mining, Interfaculty Institute of Microbiology and Infection Medicine Tübingen (IMIT), Cluster of Excellence ‘Controlling Microbes to Fight Infections’, University of Tübingen, Auf der Morgenstelle 28, 72076 Tübingen, Germany; athina.gavriilidou@uni-tuebingen.de (A.G.); nadine.ziemert@uni-tuebingen.de (N.Z.); 7German Center for Infection Research (DZIF), Partner Site Tübingen, 72076 Tübingen, Germany; 8Department of Bioresources for Bioeconomy and Health Research, Leibniz Institute DSMZ-German Collection of Microorganisms and Cell Cultures, Inhoffenstraße 7B, 38124 Braunschweig, Germany; 9Department of Microbiology, Technical University of Braunschweig, 38124 Braunschweig, Germany

**Keywords:** Indonesia, biodiversity, novel antibiotics, drug screening, bioactivity, gene cluster networking, GNPS

## Abstract

Indonesia is one of the most biodiverse countries in the world and a promising resource for novel natural compound producers. Actinomycetes produce about two thirds of all clinically used antibiotics. Thus, exploiting Indonesia’s microbial diversity for actinomycetes may lead to the discovery of novel antibiotics. A total of 422 actinomycete strains were isolated from three different unique areas in Indonesia and tested for their antimicrobial activity. Nine potent bioactive strains were prioritized for further drug screening approaches. The nine strains were cultivated in different solid and liquid media, and a combination of genome mining analysis and mass spectrometry (MS)-based molecular networking was employed to identify potential novel compounds. By correlating secondary metabolite gene cluster data with MS-based molecular networking results, we identified several gene cluster-encoded biosynthetic products from the nine strains, including naphthyridinomycin, amicetin, echinomycin, tirandamycin, antimycin, and desferrioxamine B. Moreover, 16 putative ion clusters and numerous gene clusters were detected that could not be associated with any known compound, indicating that the strains can produce novel secondary metabolites. Our results demonstrate that sampling of actinomycetes from unique and biodiversity-rich habitats, such as Indonesia, along with a combination of gene cluster networking and molecular networking approaches, accelerates natural product identification.

## 1. Introduction

It is now 80 years ago that Selman Waksman and Boyd Woodruff discovered actinomycin from *Actinomyces (Streptomyces) antibioticus*, which was the first antibiotic that was isolated from an actinomycete [1]. Since then, actinomycetes have been widely used as sources for drug discovery and development [2]. Most antibiotics and other useful natural products applied in human medicine, veterinary, and agriculture are derived from these filamentous bacteria [3,4]. Within the family of Actinomycetales, *Streptomyces* is the most prominent genus in respect to the production of bioactive secondary metabolites since it is the origin of more than 50% of all clinically useful antibiotics [5]. Successfully, the intensive screening campaigns of soil-derived streptomycetes yielded many currently recognized drugs, such as the antibacterial substance streptomycin, the antifungal metabolite nystatin, and the anticancer compound doxorubicin during the golden era of antibiotics [6,7]. However, in the last few decades, discovering and developing new drugs from these soil microorganisms has declined immensely, while the need for new drugs to overcome multidrug resistance has become greater than ever [8]. Nowadays, one of the major problems in antibiotic screening programs, in particular with streptomycetes, is the high rediscovery rate of already-known antibacterial compounds through the classical bioactivity-guided paradigms [3].

Sampling actinomycetes from conventional environments such as soils often leads to the rediscovery of known species producing already-known antibiotics [9]. Thus, gaining access to unusual unique habitats with the pursuit to isolate new strains as sources of novel bioactive compounds represents a current barrier in drug discovery research [9]. In recent years, the bioprospection of underexplored niches such as extreme or marine environments has become an efficient approach to find novel *Streptomyces* species that might produce novel compounds [10,11]. *S. asenjonii* strain KNN 42.f, isolated from a desert soil sample, is one example of a novel *Streptomyces* species from an extreme habitat, which produces the three new bioactive compounds asenjonamides A–C [12]. Another example displays the marine *S. zhaozhouensis* CA-185989 that produces three new bioactive polycyclic tetramic acid macrolactams [13]. *Micromonospora* sp. as turbinimicin producer represents a further example of prolific marine bacteria that can deliver new antifungal compounds [14]. These are only a few examples demonstrating that unusual or aquatic territories can be promising avenues as new natural products reservoirs.

Indonesia is the world’s largest archipelagic country, spanning into three time zones, covering more than 17,000 islands, with 88,495,000 hectares of tropical forest, 86,700 square kilometers of coral reefs, and 24,300 square kilometers of mangrove areas [15,16]. It has the second-highest level of terrestrial biodiversity globally after Brazil [17], while being ranked as first if marine diversity is taken into account [16,17]. With the given species-rich flora and fauna besides endemic and ecologically adapted species, mega biodiversity of microbial species is gratifyingly represented across various unique habitats [18,19,20], such as acidic hot springs [21], peatland forests [22], the Thousand Islands reef complex [23], Enggano Island [24], fish species [25], and leaves of traditional medicinal plants [26]. Thus, since unique Indonesian niches are expected to deliver untapped potential actinomycetal strains that may produce novel bioactive secondary metabolites, different locations were targeted for the sampling of actinomycetes in this study.

The latest analyses of genome sequence data from actinomycetes revealed a remarkable discrepancy between the genetic potential of the secondary metabolism, known to be encoded by biosynthetic gene clusters (BGCs), and the actual natural compound production capacity of such isolates, upon their growth under standard laboratory conditions. This is attributed to the fact that numerous BGCs are not expressed under conventional lab parameters and occur as so-called “silent” or “cryptic” BGCs [27]. The activation of these silent clusters allows one to unlock the chemical diversity of the tested organisms and enables the discovery of new molecules for medical and biotechnological purposes [28]. Thus, several efforts, e.g., involving genetic and cultivation methods, are employed to activate the expression of silent gene clusters [29]. One cultivation-based approach to exploit the metabolic capacity of the natural compound producers is the “one strain many compounds (OSMAC)” strategy [3,28,30]. Such a strategy simply relies on the variation of media compositions as a basis to test for different natural compound production profiles since global changes in the specialized metabolic pathways can occur under variable cultivation conditions [31]. The OSMAC concept represents a well-established model that was suggested nearly two decades ago; however, it still leads to the discovery of new chemotypes, such as the novel aromatic polyketide lugdunomycin from *Streptomyces* sp. QL37 [32] or an eudesmane sesquiterpenoid and a new homolog of the Virginiae Butanolides (VB-E) from from *Lentzea violacea* strain AS 08 [33]. Along the lines of the OSMAC concept, an elicitor screening approach has recently been suggested, which intends to mimic natural trigger molecules that can induce the biosynthesis of formerly unknown metabolites. This format has been conducted in a high-throughput approach and was coupled with MALDI-MS analysis. In the case of *S. ghanaensis*, this strategy led to the discovery of the antibiotically active depsipeptide cinnapetide [34].

Besides the variable trials to elicit the BGCs via pleiotropic approaches, a mass spectrometry dereplication step is frequently included in the current screening programs to address the formerly stated challenge of the high rediscovery rate prior to the tedious screening, isolation, and purification processes [35,36,37]. The utility of such a platform is to pinpoint known compounds in the initial phase of the discovery pipeline and leverage the process of finding new drugs. Integrated genomic and metabolomic mining methods have proven as an efficient dereplication strategy for compound identification in recent years [38,39,40,41]. While genome mining involves the identification of putative BGCs based on the genome sequences of the natural compound producers [42,43] using in silico bioinformatics tools such as antiSMASH [44], metabolome mining encompasses sorting out the chemical compounds in extracts of natural compound producers via their mass fragmentation patterns. Counting on the fact that metabolites with a similar chemical architecture tend to generate similar mass fragmentation patterns in mass spectrometry (MS) analysis, the implementation of the computational platform Global Natural Product Social (GNPS) to group the structurally related entities, often derive from a common biosynthetic origin, as a connected set of a molecular family cluster is an overgrowing necessity [45]. Such a platform iteratively proves its effectiveness to arrange seamlessly large numbers of samples enabling dereplication and tentative structural identification and/or classification [46]. The combinatorial employment of both computational tools side by side empowers the rapid identification of new substances, which can be highlighted by discovering the antibacterial substance thiomarinol from *Pseudoalteromonas luteoviolacea* [38] and microviridin 1777, a chymotrypsin inhibitor from *M. aeruginosa* EAWAG 127a [47].

Taken all together with the promises that highly biodiverse habitats can offer in synergy with an effective and practical mining technique, this study aimed to characterize the secondary metabolomes of selected actinomycetes isolated from three different locations within Indonesia. A collection of 422 actinomycetes from Lombok, Bali, and Enggano Islands were sampled and preliminary filtered with different bioactivity tests, where nine actinomycetes with the most bioactive potential were nominated for a hybrid genome mining and molecular networking approach in order to assess their biosynthetic capacity for the production of novel natural compounds.

## 2. Results and Discussion

### 2.1. Isolation and Characterization of Indonesian Actinomycetes

To isolate actinomycetes, soil samples were collected from two specific habitats (terrestrial and marine) in three different geographic areas of Indonesia using standard isolation protocols [48,49,50,51,52]. Enggano Island was chosen as a sampling location for terrestrial habitats since it is a pristine island with many endemic species and high biodiversity [53,54], whereas Bali and Lombok Island were selected as sampling sites for marine habitats resulting in 422 strains in total (Table 1). Among all sampling locations, the Enggano Island soil samples contributed to the highest number of actinomycetes isolates (56.2%), followed by sediment samples from Lombok (37.2%) and Bali island (6.6%).

**Table 1 marinedrugs-19-00316-t001:** Indonesian strains, isolation method, source of isolation (compare Figure 1), and most closely related species (%) based on 16S rRNA gene sequence phylogenetic analysis with EzTaxon.

Strain (*Streptomyces* sp.)	Isolation Method	Source of Isolation	Most Closely Related Species Based on 16S rDNA Analysis
SHP 22-7	phenol	Soil under a ketapang tree (*Terminalia catappa*) from Desa Meok (B1), Enggano Island	*Streptomyces rochei* NBRC 12908^T^ (99.59%)
SHP 20-4	phenol	Soil under a kina tree (*Cinchona* sp.), Desa Banjarsari (B2), Enggano Island	*Streptomyces hydrogenans* NBRC 12908^T^ (99.68%)
SHP 2-1	phenol	Soil under a hiyeb tree (*Artocarpus elastica*) near Bak Blau water spring, Desa Meok (B3), Enggano Island	*Streptomyces griseoluteus* NBRC 13375^T^ (98.96%)
DHE 17-7	dry heat	Soil under a ficus tree (*Ficus* sp.), Desa Boboyo (B4), Enggano Island	*Streptomyces lannensis* TA4-8^T^ (99.78%)
DHE 12-3	dry heat	Soil under a cempedak tree (*Artocarpus integer*), Desa Boboyo (B4), Enggano Island	*Streptomyces coerulescens* ISP 51446^T^ (98.87%)
DHE 7-1	dry heat	Soil under a terok tree (*Artocarpus elastica*), desa Boboyo (B4), Enggano Island	*Streptomyces adustus* WH-9^T^ (99.59%)
DHE 6-7	dry heat	Soil under forest snake fruit tree (*Salacca* sp.), Desa Malakoni (B5), Enggano Island	*Streptomyces parvulus* NBRC 13193^T^ (98.55%)
DHE 5-1	dry heat	Soil under a banana tree (*Musa* sp.), Desa Banjar sari (B2), Enggano Island	*Streptomyces parvulus* NBRC 13193^T^ (99.79%)
BSE 7-9	NBRC medium 802	Mangrove sediment near plant rhizosphere, Kuta (C1), Bali Island	*Streptomyces bellus* ISP 5185^T^ (99.06%)
BSE 7F	NBRC medium 802	Mangrove sediment near plant rhizosphere, Kuta (C1), Bali Island	*Streptomyces matensis* NBRC 12889^T^ (99.72%)
I3	humic acid-vitamin + chlorine 1%	Mangrove sediment from Pantai Tanjung Kelor, Sekotong (D2), West Lombok Island	*Streptomyces longispororuber* NBRC 13488^T^ (99.23%)
I4	humic acid-vitamin + chlorine 1%	Mangrove sediment from Pantai Tanjung Kelor, Sekotong (D2), West Lombok Island	*Streptomyces griseoincarnatus* LMG 19316^T^ (99.89%)
I5	humic acid-vitamin + chlorine 1%	Mangrove sediment from Pantai Tanjung Kelor, Sekotong (D2), West Lombok Island	*Streptomyces viridodiasticus* NBRC13106^T^ (99.31%)
I6	humic acid-vitamin + chlorine 1%	Mangrove sediment from Pantai Tanjung Kelor, Sekotong (D2), West Lombok Island	*Streptomyces spongiicola* HNM0071^T^ (99.78%)
I8	humic acid-vitamin	Sea sands from Pantai Koeta (D3), Lombok Island	*Streptomyces smyrnaeus* SM3501^T^ (98.44%)
I9	humic acid-vitamin	Sea sands from Pantai Koeta (D3), Lombok Island	*Streptomyces gancidicus* NBRC 15412^T^ (98.82%)

Within the frame of a preliminary bioactivity screening, all 422 isolates were evaluated for their antimicrobial activities in agar plug diffusion bioassays against selected Gram-positive (*Bacillus subtilis, Micrococcus luteus*, and *Staphylococcus carnosus*) and Gram-negative bacteria (*Escherichia coli* and *Pseudomonas fluorescens*). The 16 most potent isolates were selected based on their antimicrobial activity against the tested organisms, indicated by the largest inhibition zones around the agar plug. All 16 isolates showed bioactivity against the Gram-positive test organism *B. subtilis* (Figure 2A), and nine exerted further activity against Gram-negative test strains (Figure 2B), while only four strains (BSE 7–9, BSE 7F, I3, and I6) displayed potency against both (Figure 2).

To investigate the phylogenetic relationship of the 16 bioactive actinomycetal isolates, 16S rRNA gene sequence analyses were performed. For this purpose, the genomic DNA was isolated from each and was used as a template in a PCR approach with 16S rRNA gene-specific primers. The resulting 16S rRNA gene amplifications were sequenced, and the 16S rRNA gene sequences were compared using the EzTaxon database (www.ezbiocloud.net/, accessed on 28 May 2018) to determine the phylotype of the strains [55]. EzTaxon analysis revealed that all isolates belong to the genus *Streptomyces* with similarity values amongst the various predicted related species ranging from 98.44–99.89% (Table 1).

Subsequently, nine strains were prioritized based on their bioactivity profile and taxonomic position. Strains SHP 22-7, BSE 7-9, BSE 7F, I3, I4, I5, and I6 were selected since they showed antibacterial activity against Gram-positive and Gram-negative bacteria (Figure 2A,B). DHE 17-7 and DHE 7-1 were selected as they exerted bioactivity against at least two different Gram-positive test strains. DHE 6-7 and DHE 5-1, which showed bioactivity against all Gram-positive test strains, were not chosen for further analysis because both strains showed a close phylogenetic relationship to *Streptomyces parvulus* (Table 1), which is a known producer of the polypeptide antibiotic actinomycin D [56]. In an initial attempt with HPLC-MS analysis of the methanolic extracts of culture samples from DHE 6-7 and DHE 5-1, actinomycin D was detected as a product (Appendix A), ruling out both strains from further investigations.

### 2.2. Phylogenomic Analysis of Nine Prioritized Indonesian Streptomyces Strains

To obtain a better understanding of the phylogenetic relationship about the prioritized nine *Streptomyces* strains, a phylogenetic analysis based on their full-length genomes sequences was performed. For this purpose and the genome mining studies mentioned below, the genomic DNA was isolated from each sample and sequenced by using the Pacific Biosciences RS II (PacBioRSII) platform [57,58,59]. The resultant genome sequences ranged in sizes between 7.05 Mbp (*Streptomyces* sp. I6) and 8.36 Mbp (DHE 17-7) and GC contents between 72.08% (DHE 7-1) and 72.47% (*Streptomyces* sp. I6) (Appendix A), which share comparable values reported for *Streptomyces* species (genome sizes of 6-12 Mb [60] and GC contents of 72–73% [61,62]).

In order to run a whole-genome phylogenetic analysis, the genome sequences were submitted to the Type (Strain) Genome Server (TYGS) (https://tygs.dsmz.de, accessed on 13 December 2019) [63], which allows a phylogenetic analysis based on full-length genome sequences and compares genomic data with the database genomes. The resulting phylogenetic information is more authentic than those obtained from 16S rDNA- or multi-locus sequence analysis (MLSA)-based classifications, which only use small sequence fragments as a basis for sequence comparisons [63]. The TYGS analysis provides information on the similarity of a strain to its nearest related type strain, derived from the digital DNA-DNA hybridization (dDDH) values calculated by the genome-to-genome distance calculator (GGDC) 2.1 (http://ggdc.dsmz.de, accessed on 13 December 2019) [64]. TYGS phylogenomic analysis revealed that all nine isolates belong to the genus *Streptomyces*. The dDDH values between the nine Indonesian strains and their closest relatives ranged between 31.4% (*Streptomyces* sp. I4) and 51.5% (*Streptomyces* sp. I6) (using GGDC distance formula *d4*) (Table 2), which is below the threshold of 70% used for species delineation [65,66], proposing a novel collection of *Streptomyces* species.

According to the TYGS phylogenomic tree, the terrestrial Enggano Island strains SHP 22-7 and DHE 17-7 belong to the same clade (clade A) (Figure 3) and most likely resemble the same type of species with a dDDH value of 86.7% (Table 2). Both bacteria are found to be closely related to *S. luteus* TRM 45540, isolated from a soil sample from China [67]. All mangrove isolates originating from sediments of Lombok Island (*Streptomyces* sp. I3, I4, and I5) and Bali Island (BSE 7F, and BSE 7-9) were allied in clade B, suggesting a correlative connection (Figure 3). Additionally, the dDDH analysis showed that BSE 7F is closely related to BSE 7-9 with a value of 95.7% and thus most likely represent the same subspecies (Table 2), while *Streptomyces* sp. I3 and I4 probably represent the same species having a dDDH score of almost 100% (Table 2). The nearest related type strain of all five mangrove strains is *S. capillispiralis* DSM 41695 isolated from a Sweden soil sample [68].

By contrast, the soil sample DHE 7-1 and mangrove *Streptomyces* sp. I6 were found to group separately in distinct clades (clade C and D, respectively) (Figure 3). The soil *S. bungoensis* DSM 41781 collected in Japan [69] shares a dDDH value of 32.3% as the closest related strain to DHE 7-1 (Table 2), while the nearest related neighbor of *Streptomyces* sp. I6 is *S. spongiicola* HNM0071, isolated from a marine sponge collected from China [70] with a dDDH value of 51.5% (Table 2). Additional information on the specific polyphasic characteristics of the representative type strains from each clade can be found in the Appendix A. Altogether, 16S rRNA gene-based phylogenetic and phylogenomic studies revealed that all nine prioritized isolates belong to the genus *Streptomyces* and, based on dDDH analysis, represent novel species (Figure 3).

### 2.3. Genetic Potential for Secondary Metabolite Biosynthesis of Nine Indonesian Streptomyces Strains

To infer the genetic potential of the strains for the biosynthesis of secondary metabolites, the genomes were analyzed bioinformatically using the web tool antiSMASH version 5.0 (https://antismash.secondarymetabolites.org, accessed on 13 November 2019) [44]. The antiSMASH analysis yielded a sum of 206 potential BGCs for the nine isolates (Table 3) with the lowest BGC count of 17 for strain *Streptomyces* sp. I3 and the highest number of 30 BGCs for strain DHE 17-7 (Table 3). On average, this makes 23 BGCs per strain, which is lower than the average value of 40 BGCs reported for *Streptomyces* genomes [71]. However, the lower BGC count is most likely a result of the underlying PacBio genome sequences, which generally yield less contigs than other sequencing technologies, resulting in less interrupted BGCs and thus less BGC counts in antiSMASH analyses. The genome of DHE 17-7 exhibited a slight correlation between genome size (8.4 Mbp) and the observable number of BGCs (30 BGCs) (Table 3 and Appendix A). Several of the identified BGCs from the nine Indonesian isolates showed a high similarity (>60%) to already-known BGCs (Figure 4), e.g., all strains harbored BGCs encoding compounds that are commonly produced by streptomycetes, such as desferrioxamine, which is a vital siderophore for the growth and development [72], hopene, as a substance of the cytoplasmic membrane modulating membrane fluidity and stability [73], and a spore pigment for protection against UV radiation [74]. This result is consistent with previous observations, where these BGCs have been reported for most analyzed *Streptomyces* genomes [71]. Ectoine and geosmin BGCs were found in all Indonesian isolates except for *Streptomyces* sp. I6 (Figure 4). Moreover, albaflavenone BGCs were uncovered in all strains, excluding *Streptomyces* sp. I6 and DHE 7-1. Interestingly, in the genomes of the mangrove-derived isolates BSE 7F, BSE 7-9, I3, I4, and I5, two ectoine BGC were identified, suggesting that the additional ectoine BGCs may play a role in the adaptation of these organisms to the osmotic stress of such high-salinity environments.

Aborycin and alkylresorcinol gene clusters were discovered in the five mangrove *Streptomyces* strains, whereas amicetin, candicidin, coelichelin, and fluostatin M-Q BGCs were only detected for the soil-based isolates SHP 22-7 and DHE 17-7. Candicidin, as an example of a fungizide [75], is most likely produced by terrestrial streptomycetes in order to defend themselves against local fungal competitors. Coelichelin is a further spotted siderophore which might be necessary for the soil-living streptomycetes to sequester poorly soluble environmental Fe^3+^ [76], which is quite scarce and highly contested by other microorganisms in soils. The discovery of the same BGC composition in strains derived from the same habitat, such as soil or mangroves, is probably attributed to the fact that each biosynthetic product has its specific biochemical relevance in the respective environment. Of the nine strains, only *Streptomyces* sp. I6 harbored a staurosporine, scabichelin, echinomycin, flaviolin, and tirandamycin BGC. Likewise, DHE 7-1 together with *Streptomyces* sp. I6 were the only representatives comprising an isorenieratene BGC among the nine strains (Figure 4). Both strains, I6 and DHE 7-1, were found to be phylogenetically distant from the other strains (Figure 3), outlining that phylogenetically related isolates tend to have similar biosynthetic elements known as BGCs shaped by the environmental conditions. A similar finding has already been made by Meij et al., who reported that ecological conditions play an important role in controlling the formation of secondary metabolites in actinomycetes [77].

To glean a more detailed picture about the BGC distribution amongst the strains, the genome sequences from the nine strains have been analyzed using the BiG-SCAPE software (https://bigscape-corason.secondarymetabolites.org/, accessed on 13 November 2019) [78]. BiG-SCAPE allows fast computation and visual exploration of BGC similarities by grouping BGCs into gene cluster families (GCF) based on their sequences and Pfam protein families similarities [79]. Comparing all shared BGCs within the nine Indonesian strains with BiG-SCAPE allows visualization of the more common BGCs (large nodes) and the less frequent ones (doubletons, (singletons are not shown) (Figure 5). With this approach, we visualized the occurrence of eight GCFs with a similarity of less than 60% similarity to known BGCs as predicted by antiSMASH. Ectoine-butyrolactone-NRPS-T1PKS GCF, which has similarities with polyoxypeptin (48%) or aurantimycin A (51%), was distributed among strains I3, I4, I5 and BSE 7F (Figure 5, Appendix A). A type III polyketide (T3PKS) GCF was shared amongst the strains DHE 17-7, SHP 22-7, and DHE 7-1, which showed 7–8% BGC similarity to the herboxidiene BGC (Figure 5, Appendix A). In the strains *Streptomyces* sp. I3 and I4 of clade B, we found the others-type I polyketide (otherks-T1PKS), which showed 48–55% BGC similarity to the nataxazole BGC, and an aminoglycoside/aminocyclitol (amglyccycl) BGC type, which led to 2% similarity to the BGC of cetoniacytone A (Figure 5, Appendix A). We identified two unique GCFs in the strains BSE 7F and BSE 7-9 of clade B, namely a transAT-PKS GCF, which showed 54–58% similarity to the weishanmycin and phenazine BGC types, and did not show any similarity to any BGC in the antiSMASH database (Figure 5, Appendix A). Moreover, we detected two GCFs of an indole, which showed 23–33% BGC similarity to the 5-isoprenylindole-3-carboxylate β-D-glycosyl ester BGC and other BGC type, which do not belong to any BGCs in the antiSMASH database for the phylogenetically related species of SHP 22-7 and DHE 17-7 of clade A (Figure 5, Appendix A). Altogether, the Big-SCAPE analysis revealed eight unique GCFs, which could not be associated with known BGCs and may have the potential to encode for new substances. Furthermore, the obtained data disclosed that phylogenetically related strains derived from a similar environmental habitat tend to share similar BGC composition profiles. Inferred from this observation, one can conclude that it is worth it to make an effort to sample actinomycetes from unique environmental habitats, since this may lead to the isolation of phylogenetically unique species, which have a higher potential of producing novel natural compounds, as also previously described by Hug et al. [9].

### 2.4. Optimal Cultivation Conditions for Compound Production of Nine Indonesian Streptomyces Strains

In order to infer the biosynthetic capacity of the prioritized nine isolates in a bioactivity context, various media following the OSMAC strategy were screened to define the optimal production conditions [30,31]. For this purpose, SHP 22-7, DHE 17-7, DHE 7-1, BSE 7-9, BSE 7F, I3, I4, I5, and I6 were each grown in twelve different liquid cultivation media (SGG, YM, OM, R5, MS, TSG, NL19, NL300, NL330, NL500, NL550, and NL800 (Appendix A)), and culture samples were harvested at different time points (48, 72, 96, and 168 h). Cell cultures were extracted with ethyl acetate, concentrated in vacuo, and then re-dissolved in methanol. Methanolic extracts were tested in bioassays against a selected panel of pathogenic strains *B. subtilis, M. luteus, S. carnosus, E. coli,* and *P. fluorescens*. Samples with the largest inhibition zones in bioassay tests were defined as the ones grown under optimal cultivation conditions. For each Indonesian *Streptomyces* strain, the optimal production conditions have been defined for cultivation in liquid media (Appendix A). In addition, it is hypothesized that filamentous actinomycetes as soil organisms grow and develop better on solid nutrient substrates and that a well-grown healthy culture produces more diverse secondary metabolites [80]. Thus, to extend the probability of finding new substances by exploring the biosynthetic potential of the nine strains for secondary metabolite production, we recruited an antibiotic extraction also from solid media. For this purpose, each isolate was spread on agar plates consisting of the respective abovementioned media and incubated for 7–10 days at 28 °C until spores formed. Grown agar samples were squeezed out and concentrated. The aqueous phase of the solid medium extract was used for bioassays and further chemical analysis.

For *Streptomyces* sp. I3 and I4, the same cultivation parameters were found to be optimal. Both strains showed a promising potency upon their growth in liquid NL550 medium for 72 h on solid MS medium (Appendix A). Such similar production behavior might be ascribed to their most possible likelihood to represent the same species as suggested above. Furthermore, we found that most of the nine isolates (*Streptomyces* sp. I3, I4, I5, and I6) produced best on solid MS medium (Appendix A). In general, MS is a suitable medium for streptomycetes regarding spore isolation [81]. This would support the hypothesis that strains produce better, when they show healthy growth and development.

### 2.5. Identification of Natural Compounds from Nine Indonesian Actinomycetes

To putatively identify the specialized bioactive substances which are produced by the nine isolates under the various conditions, the culture extract samples were submitted to high-resolution mass spectrometry (HRMS) coupled with the GNPS platform. For this purpose, the obtained extracts from the optimal medium in liquid and solid were firstly fractionated by solid-phase extraction (SPE) and then qualitatively profiled against their main crudes and media controls using HPLC. Subsequently, the prioritized profiles and/or SPE fractions that mainly cover the whole metabolomes with fewer media components were chosen for further metabolomics mass identification through HRMS/MS. The acquired tandem-MS mass spectra from the positive mode were recruited to build a feature-based molecular network, while the negative ionization was consulted, if needed, during the annotation step to further validate the feature identities [45,82]. The dereplication of the known compounds, chemical analogues, and potential novel chemical structures was carried out either by matching their MS/MS spectra against the literature if available, GNPS spectral libraries [45] and/or assisted by manual in silico annotation via Sirius+CSI: FingerID 4.0.1 integrated with Antibase and Pubchem databases [83,84] (see Material and Methods).

Among the numerous identified secondary metabolites from the nine isolates, antimycins cluster were swiftly retrieved through the identical similarity of their MS/MS spectra to the publicly shared ones of GNPS libraries (Appendix A). Tracking down such features in liquid BSE 7F fractions, particularly the one eluted with 100% MeOH in negative mode, expanded this set with further known members (Appendix A). In alignment with the formerly described positional and stereogenic isomers of the antimycin family entities, the extracted ion chromatograms (EICs) unambiguously displayed such an isomeric behavior under both modes (Appendix A) [85,86,87]. In a similar fashion to antimycins, a different cluster comprising ferrioxamines was deciphered with the aid of GNPS spectral libraries. Ferroxamine D1, 656.2830 Da in size as C_27_H_48_N_6_O_9_ [88,89], was displayed as the primary ion linked with an additional unknown analogue, 627.3303 Da as C_26_H_51_FeN_8_O_6_ (Appendix A). Despite the fact of observing these two features under only solid cultivation parameters across different isolates (*Streptomyces* sp. I3, I4, I6, BSE 7F, DHE 17-7, and SHP 22-7) with variable concentrations, two extra unknown amphiphilic trihydroxamate-containing siderophores were also grouped (Appendix A). Interestingly, BSE 7-9 was the sole producer of such amphiphilic entities under exclusive liquid conditions. Moreover, two additional unknown ferrioxamines were retrieved as unique features singly produced by the DHE 17-7 isolate (Appendix A).

Analogously, staurosporine, with two further congeners, was dereplicated from the I6 sample assisted by shared spectral repositories (Appendix A). Manual annotation of a pair of singletons, 1137.45 as [M+H]^+^ and 560.22 as [M + 2H]^2+^ from the I6 extract, uniquely grown under solid conditions, could decipher echinoserine and depsiechinoserine, respectively (Appendix A) [90,91]. Although the two features were supposed to group together considering their skeletons, the MS/MS spectra of their triggered singly and doubly pseudomolecular ions were different enough not to serve such a purpose resulting in scattered self-looped nodes (Appendix A). Furthermore, traces of the structurally related echinomycin [92] were also observed within *Streptomyces* sp. I6 extracts, expanding in this way the molecular compound family (Appendix A). Likewise, a tirandamycins cluster was disclosed in *Streptomyces* sp. I6 extracts upon liquid cultivation depicting the known tirandamycin A in connectivity with further related chemotypes (Appendix A). In parallel, the observed UV absorbance of the annotated mass ion at m/z 418.18 as tirandamycin A was in alignment with its reported characteristic value [93,94], additionally confirming the identity of the dereplicated feature (Appendix A). Notably, the anticipated molecular formula of the grouped ions of the tirandamycin cluster, besides their degrees of unsaturation, was also reflected by their observed UV absorbances, which differed from the characteristic known one (Appendix A).

An additional constellation of ions mainly derived from isolates BSE 7-9 and I5 was uncovered through manual annotation as naphthyridinomycins cluster (Appendix A). The in silico annotation considering the molecular formula prediction and their MS^2^ spectra deconvoluted naphthyridinomycin-A, aclidinomycin A, and bioxalomycin-β2 besides several unknown related products (Appendix A) [95,96,97]. Similarly, the manual interrogation of an exclusive group of ions derived from DHE 17-7 led to the putative dereplication of ECO-501, a PKS product so far only reported from *Amycolatopsis orientalis* ATCC 43491 [98] (Appendix A). Interestingly, the putative annotation of such a feature was in complete alignment regarding the observed UV absorbance and the formerly reported MS/MS spectra (Appendix A). Moreover, amicetin and cytosaminomycins as structurally related entities were uncovered from SHP 22-7 samples as a big group of ions (Appendix A), encompassing a wide scope of structural modifications as expected according to previous reports in addition to a putatively new set of congeners (Appendix A) [99,100,101].

The compound naphthyridinomycin was detected in several culture extract samples from strains of mangrove origin, such as *Streptomyces* sp. I3, I4, I5, BSE 7F, and BSE 7-9 (Figure 6, Table 4), while amicetin was detected as a biosynthetic product from the isolates SHP 22-7 and DHE 17-7 obtained from soil samples of Enggano Island (Figure 6, Table 4). Furthermore, we observed that *Streptomyces* sp. I6 produces echinomycin (Figure 6, Table 4), a substance also reported as the biosynthetic product from the closely related type strain *Streptomyces spongiicola* HNM0071, which was originally derived from a marine sponge [102]. These results underline our assumption that phylogenetically related strains are likely to produce similar compounds as a response to their natural-habitat environmental conditions. Specifically, the isolates *Streptomyces* sp. I3 and I4 have been found to most likely represent the same species derived from a similar habitat as indicated by the dDDH value of almost 100% and the high overall similarities of BGC composition and secondary metabolite production profile of both strains (see above). In this context, it should be mentioned that current antibiotic research often addresses the problem of dereplication of known compounds during drug-screening approaches [103,104,105]. However, what should also be taken into account is the fact that there is also an issue of dereplication of producer strains as observed in the current study. Thus, it is worth it to put effort into phylogenetic profiling at the beginning of the screening strategy in order to sort out known producer strains.

Interestingly, the ferrioxamine molecular family was only detected for samples of strains grown on solid media (Table 4 and Table 5). In addition to the abovementioned metabolites, the solid media uniquely delivered a putative new molecular family consisting of likely three peptides with *m/z* 598.2834 [M + 2H]^2+^, 662.8048 [M + 2H]^2+^, and 727.3259 [M + 2H]^2+^, for which no known substance could be associated. These compounds were detected in samples of strains I3, I5, and BSE 7F (Appendix A, Table 5), highlighting that cultivation conditions have a substantial effect on the chemical profiles. A further example of rendering the impact of the adopted cultivation method was represented with an additional cluster of unknown features from SHP 22-7 isolate, designated compound group I, which were exclusively produced under nonliquid fermentation (Appendix A).

Within the same context, strain DHE 17-7 also offered several putative new compounds (compound group II) which were detected when grown in a liquid medium and presented themselves only as a set of doubly charged entities (Appendix A) (Table 5). Thus, in regard to drug-discovery efforts, strain DHE17-7 is the most promising strain to be investigated further. The potent biosynthetic capacity is also reflected by the genetically encoded biosynthetic potential since DHE17-7 has a total of 30 BGCs, which is the largest BGC set compared to the other Indonesian strains (Table 3). In summary, 16 potential novel compounds (Table 5) have been identified as biosynthetic products from the Indonesian strains, which could not be associated with any known compound and thus demonstrate the value of new strains for drug-discovery research.

Furthermore, we observed a correlation between growth conditions and compound production. It is known that sources of complex nitrogen such as soybean meal and corn steep liquor can increase ferrioxamine production in streptomycetes [106,107]. Interestingly, ferrioxamine B/D and its analogs has been mainly identified for strains grown on solid media, such as MS agar (*Streptomyces* sp. I3, I4, I6) and NL300 agar (SHP 22-7) (Table 4 and Appendix A), which contain soy flour and cotton seed powder, respectively (Appendix A). We could detect ferrioxamines only in samples obtained from strains grown on solid medium. This might be because in liquid media iron (Fe^3+^) is more evenly distributed compared to solid media. Thus, cells grown on solid media might be faced with local iron depletion conditions, which lead to induction of ferrioxamine biosynthesis [76]. In addition to ferrioxamine and its analogs, several known and unknown compounds were only discovered in samples from strains grown on solid medium, i.e., the three known compound echinomycin, staurosporine, and tirandamycin for *Streptomyces* sp. I6, and three putative new peptides for *Streptomyces* sp. I3; I5; BSE 7F, as well as the putative new compound group I for *S.* sp. SHP 22-7 (Table 4 and Table 5). Apart from that, we also found some unknown and known compounds in strains grown in liquid media only, such as amicetin (SHP 22-7 and DHE 17-7), antimycin and its analogs (BSE 7F), ECO-0501 (DHE 17-7), or the putative new compound group II for strains *Streptomyces* sp. DHE 17-7 (Table 4 and Table 5). This indicates that cultivation conditions significantly affect the formation of substances. Therefore, both the liquid and solid cultivation approach are feasible for increasing the probability of discovering new compounds.

### 2.6. Identification of Potential BGCs Responsible for Compound Production in the Nine Indonesian Streptomyces Strains

To identify the BGCs responsible for compound production in the nine Indonesian *Streptomyces* strains, we aimed to link the compound production profile and BGC composition by correlating the BGCs data with the MS-based molecular networking results. As described above, strains SHP 22-7, I3, I4, and I6 produce desferrioxamine B/D when grown on solid media (Table 4). We observed that the corresponding BGCs associated with desferrioxamine B/D biosynthesis were present in all of the four strains. Furthermore, we were able to assign the BGC responsible for the biosynthesis of naphthyridinomycin in the strains BSE 7F, BSE 7-9, I3, I4, and I5 (Table 4). Additional BGCs could be assigned to the compound formations of amicetin in SHP 22-7 and DHE 17-7, antimycin in BSE 7F, echinomycin, staurosporine, and tirandamycin A in I6 (Table 4). Furthermore, we could not identify the BGC encoding the biosynthesis of ECO-0501 in strain DHE 17-7 based on the antiSMASH output. A potential candidate gene cluster could be cluster region 24, which is a predicted type I PKS BGC that shows some similarity (<55%) to BGCs encoding structurally related macrolactam natural products, such as vicenistatin, sceliphrolactam, and streptovaricin (Appendix A).

In addition to the metabolites mentioned earlier, we also discovered a group of new peptides, which were detected in samples of strains *Streptomyces* sp. I3, I5, and BSE 7F grown on solid media (Table 5). Notably, for all three strains a bacteriocin BGC could be detected (Appendix A), which showed 42–57% similarity to the informatipeptin BGC. Alternatively, all three strains also share a combined NRPS/ectoine/butyrolactone/other/T1PKS gene cluster (Appendix A), and it is also conceivable that the peptide group might be encoded from this region. A similar cluster was found on regions 21 and 22 for the phylogenetically related strain BSE 7-9, for which, however, no respective compound was detected (Appendix A). Moreover, we found the putative new compound group IIwith masses ranging from 435–474 Da, produced by strain DHE 17-7 when grown in liquid medium (Table 5). Five BGCs are present in DHE 17-7 (region 10, 16, 17, 22, and 28), which do not show any similarity to known BGCs in the antiSMASH database and nine BGCs (region 3, 4, 6, 11, 13, 19, 24, 25, and 26) have similarities of less than 50%. Thus, the so far unknown metabolites might be encoded by some of the unique BGCs from DHE 17-7 (Appendix A). The same applies for the putative new compound group I detected in strain SHP 22-7. Its genome comprises 13 BGCs with similarities of less than 50% and therefore represents all putative candidates. Similar observations have been made in comparable studies, where it has been shown that BGCs encoding ectoine, desferrioxamine, spore pigment, and bacteriocin production are very abundant in actinobacterial natural compound producers; however, each strain still possesses numerous BGCs that code for potential yet unknown substances [108,109,110]. That Indonesian habitats can serve as a promising reservoir for antibiotic active substances has already been highlighted in several previous screening studies [111,112,113,114,115]. Especially Indonesian actinomycetes have been reported as producer strains of new secondary metabolites, as for example shown for the Indonesian *Streptomyces* sp. strains ICBB8230, ICBB8309, and ICBB8415, which produced new angucyclinones [116,117], *Streptomyces* sp. ICBB8198, producing new phenazine derivatives [118], and *Streptomyces* sp. ICBB9297, which produced new elaiophylin macrolides [119]. Furthermore, Indonesian non-*Streptomyces* strains, as for example *Micrococcus* sp. ICBB8177 and *Amycolatopsis* sp. ICBB8242, have also been reported to produce novel compounds, as for example the limazepines or succinylated apoptolidins, respectively [120,121]. Thus, Indonesian habitats can indeed be considered a promising source for new bioactive natural products.

Altogether, the combined GNPS and cluster networking approach disclosed several potentially novel compounds from the Indonesian strains *Streptomyces* sp. I3, I4, I5, I6, BSE 7F, BSE 7-9, and DHE 17-7—some of which could be assigned to potential encoding BGCs, and some are expected to be encoded by unique BGCs. The new Indonesian isolates thus represent a valuable resource for further drug research and development approaches. We conclude that the combined phylogenomic, GNPS, and cluster-networking approach is an efficient strategy to prioritize phylogenetically unique producer strains and focus on potentially novel compounds encoded by special BGCs.

## 3. Materials and Methods

### 3.1. Sample Collection and Treatment

Soil samples were collected from Enggano Island (5°22′57.0792″ S, 102°13′28.2792″ E), Indonesia, in December 2015 (Figure 1B). Marine samples were collected from marine sediments from Bali Island (8°43’5.5″ S, 115°10′7.8″ E), Indonesia, in May 2014 (Figure 1C), and Lombok Island West Nusa Tenggara (8°24′17.133″ S, 116°15′57.228″ E), Indonesia, in May 2017 (Figure 1D). Soil and sediment samples were taken aseptically from 10 cm depth of soil samples and the center of sediment in mangrove and tidal area. Soil and sediment samples were transferred into sterile 50 mL conical tubes and placed on ice and then stored at 4 °C until further treatment.

### 3.2. Isolation of Actinomycetes

Isolation and enumeration of actinomycetes were done using a serial dilution of Humic Acid-Vitamin (HV) medium [48] and/or NBRC No. 802 Medium [49] by using the direct method [50], the dry heat method [51], and the phenol method [51]. In the direct method, an air-dried soil sample or marine sediment was ground in a mortar and heated in a hot-air oven at 110 °C for 30 min. One gram of the heated samples was transferred to 10 mL of sterile water and mixed for 2 min, then diluted with sterile water to 10^−1^, 10^−2^, and 10^−3^ times. In total, 200 µL of each dilution was inoculated on isolation medium agar of HV [48] or NBRC No. 802 Medium [49] with or without the addition of 1% NaCl. The inoculated plates were incubated for 2–4 weeks at 28 °C. The colonies showing the *Streptomyces* morphological characteristics were selected and streaked on fresh plates of the modified *Streptomyces* International Project 2 (ISP2 ≙ YM) agar [52]. The cultures were resuspended in sterile 0.9% (*w*/*v*) saline supplemented with 15% (*v*/*v*) glycerol and stored at −80 °C. This dry-heat method [51] was used to isolate heat-tolerant actinomycetes spores. In the dry-heat method, the soil or sediment samples were incubated at 100 °C for 40 min and then cooled to 28 °C in a desiccator. The samples were distributed on HV medium agar plates with a spatula tip and incubated at 28 °C for 2–3 weeks. The phenol method was used to select for spores, which survive in the presence of phenol. In total, 1 mL of 10^−1^ dilution of one gram of oven-dried soil or marine sample was transferred to 9 mL of sterile 5 mM-phosphate buffer (pH 7.0) containing phenol at a final concentration of 1.5%. The sample was then heated and diluted in serial dilution (10^−1^, 10^−2^, 10^−3^). Next, 100 or 200 µL of each dilution was spread over the surface of HV medium agar plates and incubated for 2–4 weeks at 28 °C.

### 3.3. Antimicrobial Bioassays

The preliminary screening of actinomycetal strains for antimicrobial activity was performed using the agar plug diffusion method (see Appendix A for test plate preparation). Gram-positive (*B. subtilis* ATCC6051, *M. luteus*, and *S. carnosus* TM300) and Gram-negative bacteria (*E. coli* K12 W3110 and *P. fluorescens*) were chosen as test organisms. The isolates were spread evenly over the agar plate surface of soya flour mannitol medium (MS) (mannitol 20 g, soy flour (full fat) 20 g, agar 16 g in 1 L of distilled water) [80] and incubated for 10 days at 28 °C. Agar discs of the 10 days inoculum were cut aseptically with a cork borer (9 mm diameter) and placed on the bioassay test plate. Bioassays to determine optimal cultivation conditions in the liquid culture were examined using a disc diffusion assay against the test Gram-positive (*B. subtilis* ATCC6051, *M. luteus*, and *S. carnosus* TM300) and Gram-negative bacteria (*E. coli* K12 W3110 and *P. fluorescens*). In total, 10 µL methanolic extract obtained from liquid cultures of the actinomycetal strains was pipetted on a filter disc (6 mm) and then placed on the respective test plates. In addition, 5 µL kanamycin (50 μg/mL) was used as positive control and 10 µl methanol as a negative control.

The bioassay plates were incubated overnight at 37 °C for *B. subtilis*, *E. coli*, and *S. carnosus* and at 28 °C for *M. luteus* and *P. fluorescens* to allow for the test organisms’ growth. The antimicrobial activity of the isolates was assessed by measuring the diameter of the inhibition zone (mm) around the agar plug or the discs. All bioassay tests were carried out as three independent biological replicates.

### 3.4. Isolation of Genomic DNA and 16S rDNA Phylogenetic Analysis

For isolation of genomic DNA, the producer strains were grown for two days in 50 mL of R5 medium at 30 °C [81]. The genomic DNA was extracted and purified with the Nucleospin^®^ Tissue kit from Macherey-Nagel (catalog number 740952.50) following the standard protocol from the manufacturer. The DNA was applied as a PCR template for 16S rRNA gene amplification using polymerase chain reaction (PCR). Primers used for PCR were 27Fbac (5′-AGAGTTTGATCMTGGCTCAG-3′) and 1492Runi (5′-TACGGTTACCTTACGACTT-3′). The PCR amplicons were subcloned into the cloning vector pDrive (Qiagen) using basic DNA manipulation procedures as previously described by Sambrook et al. [122]. The respective 16S rDNA fragments were sequenced at MWG Eurofins (Ebersberg, Germany) with primers 27Fbac. The 16S rDNA sequence data were analyzed using the EzTaxon database (https://www.ezbiocloud.net, accessed on 28 May 2018).

### 3.5. Phylogenomic and Genome Mining Analysis

For phylogenomic and genome mining studies, full-genome sequence data have been obtained as reported previously [57,58,59]. Genomic DNA was isolated to construct a 10–20 kb paired-end library for sequencing by Macrogen (Seoul, South Korea) with the Pacific Biosciences RS II technology (Pacbio). The genome was assembled using Hierarchical Genome Assembly (HGAP) V.3. and annotated with Prokka version 1.12b and the NCBI Prokaryotic Genome Annotation Pipeline (PGAP). The phylogenomic analysis of the nine selected strains was carried out with the Type (Strain) Genome Server (TYGS), a free bioinformatics tool (https://tygs.dsmz.de/, accessed on 13 December 2019) for whole-genome-based taxonomic analysis [63]. The identification of potential biosynthesis gene clusters (BGCs) was accomplished by analyzing the genome sequences with antiSMASH version 5.0 [44]. The antiSMASH results were further analyzed using the BiG-SCAPE platform [78] to cluster the predicted BGCs into gene cluster families (GCFs) based on their sequences and Pfam protein family similarities [79]. BiG-SCAPE was conducted on global mode with default parameters [78], with the exception of the raw distance cutoff and the “--mix”parameter. Raw distance cutoff was set to 0.4 to ensure that even clusters with a pairwise distance higher than 0.3 (the default) were included in the output. The resulting network of BiG-SCAPE was visualized with Cytoscape version 3.7.2 [82].

### 3.6. Cultivation Conditions for Optimal Compound Production of Nine Indonesian Strains

To determine optimal cultivation conditions in liquid culture, the nine Indonesian actinomycetes strains, SHP 22-7, DHE 17-7, DHE 7-1, BSE 7-9, BSE 7F, I3, I4, I5, and I6, were each cultivated in 50 mL inoculum medium (NL410) in 500-mL Erlenmeyer flasks (with steel springs) in an orbital shaker (180 rpm) at 28 °C. After 48 h, 10 mL of preculture was inoculated into 100 mL of twelve different production medium (SGG, YM, OM, R5, MS, TSG, NL19, NL300, NL330, NL500, NL550, and NL800 (Appendix A) and cultivated for 48–168 h. Cell culture samples were harvested at different time points (48, 72, 96, and 168 h). In addition, 5 mL of each cell culture sample was extracted with the same volume of ethyl acetate (EtOAc) for 30 min at room temperature. The EtOAc was dried in a rotary evaporator and suspended in a total volume of 0.75 mL methanol. The methanolic extracts were used for bioassay experiments. The culture extract samples, which yielded the largest zone of inhibition in the bioassays against the test organisms, were used for further compound identification analysis. To determine optimal cultivation conditions on solid culture, the nine Indonesian strains were each spread on 100 mL agar plates consisting of the respective abovementioned cultivation media and then incubated for 7–10 days at 28 °C until spore formation was visible on agar plates. The overgrown agar was then used for bioassay experiments and further compound identification analysis.

### 3.7. Sample Preparation for Chemical Identification

For chemical identification in the liquid sample, the nine Indonesian isolates were each cultivated in 50 mL of NL410 medium. After 48 h, 10 mL of the preculture was inoculated into 100 mL of optimal production medium. The 100 mL whole broth of each cell culture was extracted as described above. Then, the extracts were used for further experiment. For chemical identification from samples grown on solid medium, overgrown agar was cut into pieces and transferred to 50 mL Falcon tubes. The Falcon tubes were centrifuged at 13,000 rpm for 30 min at room temperature. The aqueous phase was concentrated to 1/5 of the original volume in the Genevac Centrifugal Evaporator EZ-2 Elite (SP Scientific). The concentrated aqueous phase was used for further chemical profiling.

The culture extract samples obtained from liquid medium extraction and the aqueous phase of the solid medium extraction were separated by solid-phase extraction (SPE) columns. The columns were washed twice with 2 mL methanol and 2 mL distilled water for activating the columns. The samples were prepared by adding 100% methanol to the culture extract samples and the aqueous phase until the samples were dissolved completely. The methanolic samples were applied onto the activated columns with a flow rate of 2 mL/min. The column was washed twice with distilled water. The column was eluted consecutively with 2 mL of 100% methanol, 50% methanol, and distilled water. Samples from the elution column were defined as fractions. The column was eluted with 100% methanol as the 100% fraction, with 50% methanol as the 50% fraction, and distilled water as the distilled water fraction. The fractions were dried in the Genevac EZ-2 Elite (SP Scientific) and then dissolved with 0.5 mL methanol. The crude extracts and all fractions were analyzed with HPLC and high-resolution mass spectrometry (HRMS).

### 3.8. HPLC-HRMS/MS Analysis

The HRMS analysis was carried out on MaXis 4G instrument (Bruker Daltonics, Bremen, Germany) coupled to an Ultimate 3000 HPLC (Thermo Fisher Scientific, Bremen, Germany). HPLC-method was applied as follows: the spectrometer using a gradient (solvent A: 0.1% formic acid (FA) in H_2_O, and solvent B: 0.06% formic acid in acetonitrile), a gradient of 10–100% B in 45 min, 100% B for an additional 10 min, using a flow rate of 0.3 mL/min; 5 μL injection volume and UV detector (UV/VIS) wavelength monitoring at 210, 254, 280, and 360 nm. The separation was carried out on a Nucleoshell 2.7 μm 150 × 2 mm column (Macherey-Nagel, Düren, Germany), and the range for MS acquisition was *m/z* 100–1800. A capillary voltage of 4500 V, nebulizer gas pressure (nitrogen) of 2 (1.6) bar, ion source temperature of 200 °C, the dry gas flow of 9 (7) l/min source temperature, and spectral rates of 3 Hz for MS1 and 10 Hz for MS^2^ were used. For acquiring MS/MS fragmentation, the ten most intense ions per MS1 were selected for subsequent collision-induced dissociation (CID) with stepped CID energy applied. The employed parameters for tandem MS were applied as previously detailed by Garg et al. in 2015 [123]. Sodium formate was used as an internal calibrant and Hexakis (2,2-difluoroethoxy) phosphazene (Apollo Scientific Ltd., Stockport, UK) as the lock mass. Data processing was performed using Bruker Daltonics Data Analysis 4.1(Bremen, Germany).

### 3.9. MS/MS Molecular Networking

Mass-spectral data were analyzed using Compass Data Analysis 4.4 (Bruker Daltonik, Bremen, Germany), whereas MetaboScape 3.0 (Bruker Daltonik, Bremen, Germany) was consulted for molecular features selection. Raw data files were imported into MetaboScape 3.0 for the entire data treatment and preprocessing in which T-ReX 3D (time-aligned region complete extraction) algorithm is integrated for retention time alignment with an automatic detection to decompose fragments, isotopes, and adducts intrinsic to the same compound into one single feature. All the harvested ions were categorized as a bucket table with their corresponding retention times, measured *m/z*, molecular weights, detected ions, and their intensity within the sample. The Bucket table was prepared with an intensity threshold (1e3) for the positive measurements with a minimum peak length 3, possessing a mass range of 150–1800 Da. For detailed parameters employed for the MetaboScape analysis, see Appendix A. The features list of the preprocessed retention time range was exported from MetaboScape as a single MGF file, which was in turn uploaded to the GNPS online platform where a feature-based molecular network (FBMN) was created. The precursor ion mass tolerance was set to 0.03 Da and a MS/MS fragment ion tolerance of 0.03 Da. A network was then created where edges were filtered to have a cosine score above 0.70 and more than 5 matched peaks. Further, edges between two nodes were kept in the network if and only if each of the nodes appeared in each other’s respective top 10 most similar nodes. Finally, the maximum size of a molecular family was set to 100, and the lowest-scoring edges were removed from molecular families until the molecular family size was below this threshold. Cytoscape 3.5.1 was used for molecular network visualization.

## 4. Conclusions

In this study, we report on the isolation of 422 actinomycetes strains from three different unique areas in Indonesia. A combined genomics and metabolomics approach was applied to nine of the most potent antibiotic producer strains, which allowed us to uncover 16 so far unknown compounds. When cultivating the strains in various liquid and solid media, we found that culture conditions significantly affected the ability to produce specific compounds. Thus, the combination of both cultivation methods, solid and liquid cultivation, is a suitable approach to tap the full biosynthetic potential of actinomycetes. By phylogeny-associated genome mining studies, we found that phylogenetically related species tend to have a similar BGC composition. Additional metabolomics data suggested that the ability of the strains to produce certain compounds may be influenced by the environmental conditions, where the producer strains have been derived from.

Overall, the described methodology represents an efficient strategy for drug discovery and the reported unknown compounds may serve as a basis for further drug development.

## Figures and Tables

**Figure 1 marinedrugs-19-00316-f001:**
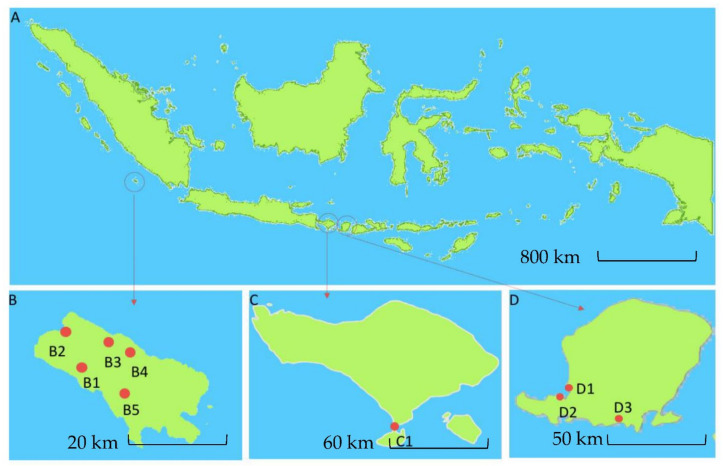
Map of Indonesia showing three geographical regions (**A**). Sampling site location in Enggano Island (**B**), Bali Island (**C**), and Lombok Island (**D**). Red dot shows the sampling locations at Enggano Island, B1: Desa Meok; B2: Desa Banjar Sari; B3: Bak Blau Waterspring, Desa Meok; B4: Desa Boboyo; B5: Desa Malakoni; at Bali Island C1 for Kuta; and Lombok Island D1: Pantai Cemara, Lembar; D2: Pantai Tanjung Kelor, Sekotong; D3: Pantai Koeta.

**Figure 2 marinedrugs-19-00316-f002:**
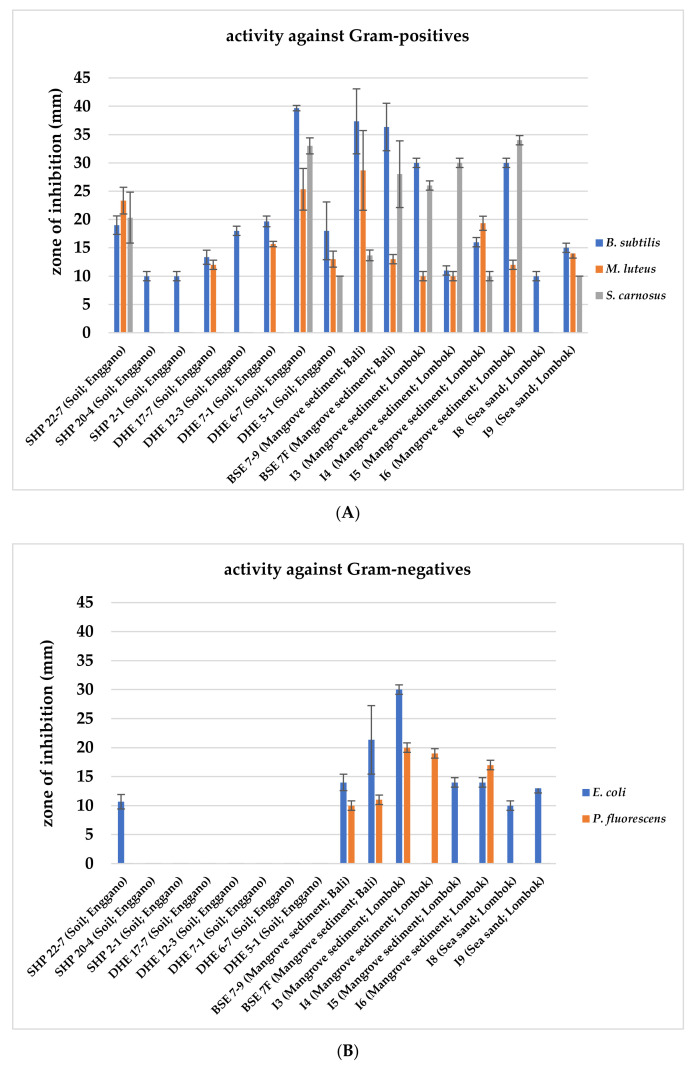
Antimicrobial bioassays with 16 Indonesian actinomycetes strain samples against Gram-positive (**A**) and Gram-negative test strains (**B**). Inhibition zone diameters of agar plug test assays are given in mm. Agar plugs were used after ten days of growth of the respective actinomycetes strains. Data shown are as the result of three independent biological replicates.

**Figure 3 marinedrugs-19-00316-f003:**
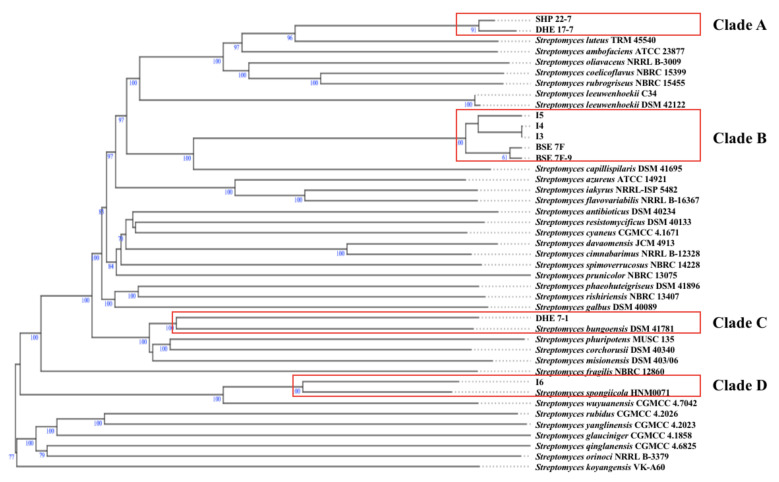
Whole-genome sequence tree generated with the TYGS web server for nine Indonesian *Streptomyces* isolates (highlighted by red boxes) and closely related type strains. Tree inferred with FastME from GBDP distances was determined from genome sequences. The branch lengths are scaled in terms of GBDP distance formula *d*_5_. The numbers above branches indicate GBDP pseudobootstrap support values > 60% from 100 replications, with an average branch support of 84.4%. The tree was rooted at the midpoint.

**Figure 4 marinedrugs-19-00316-f004:**
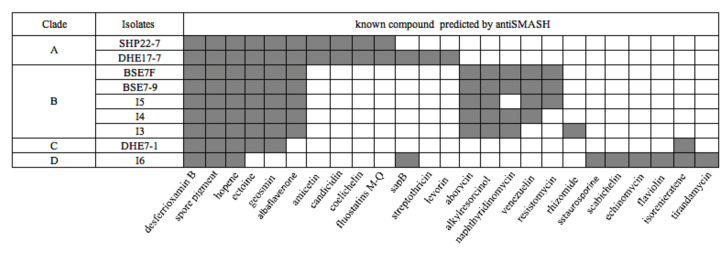
Presence (grey color) and absence (white color) of BGCs in nine Indonesian strains as predicted by antiSMASH analysis with similarity above 60%.

**Figure 5 marinedrugs-19-00316-f005:**
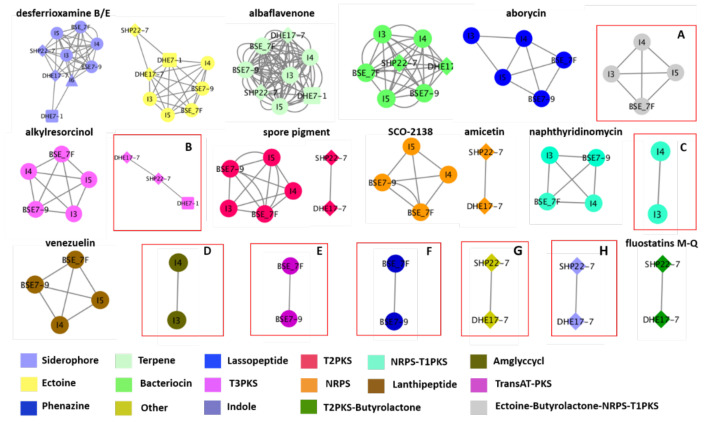
Similarity network of the predicted biosynthetic gene clusters (BGCs) of the nine Indonesian *Streptomyces* strains. Shared similar BGCs are indicated by a connected line. Each node represents a specific BGC type (labeled with different colors). The shape node represents the same species, i.e., clade A (SHP 22-7 and DHE 17-7) indicated with diamond, clade B (I4, I5, BSE 7F, and BSE 7-9) shown with ellipse, clade C (DHE 7-1) with a cube, and clade D (I6) indicated with a triangle. BGCs with similarities less than 60% are highlighted by red boxes: (**A**) Ectoine-butyrolactone-NRPS-T1PKS; (**B**) T3PKS; (**C**) Otherks-T1PKS, (**D**) Amglyccyc; (**E**) TransAT-PKS; (**F**) Phenazine; (**G**) Other; and (**H**) Indole.

**Figure 6 marinedrugs-19-00316-f006:**
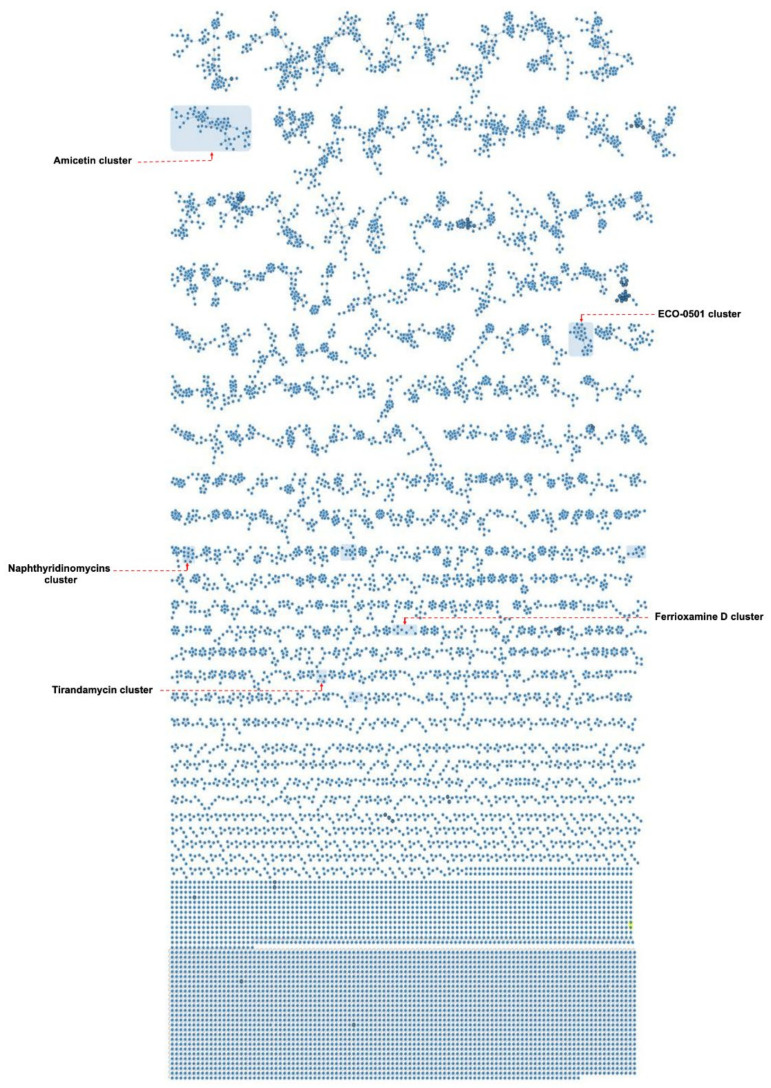
Molecular networking of extract and fraction samples from nine Indonesian *Streptomyces* strains. Molecular families containing a known substance are highlighted by blue boxes.

**Table 2 marinedrugs-19-00316-t002:** Data from pairwise comparisons between genome sequences from nine Indonesian strains and their closest related strains based on dDDH analysis. “Query strain” refers to analyzed strain, and “subject strain” refers to most closely related Indonesian strain sample. Degree of relatedness is given as dDDH distance formula *d4* as previously described by Meyer-Kolthoff et al. [66].

Query Strain	Subject Strain	dDDH (*d*_4_, in %)
I3	I4	99.6
BSE 7-9	BSE 7F	95.7
DHE 17-7	SHP 22-7	86.7
I4	I5	82.6
I3	I5	82.5
BSE 7F	I5	78.4
BSE 7-9	I5	78.4
BSE 7-9	I4	77.2
BSE 7F	I4	77.2
BSE 7F	I3	77
BSE 7-9	I3	77
I6	*Streptomyces spongiicola* HNM0071	51.5
SHP 22-7	*Streptomyces luteus* TRM 45540	43.6
DHE 17-7	*Streptomyces luteus* TRM 45540	40.3
DHE 7-1	*Streptomyces bungoensis* DSM 41781	32.3
I3	*Streptomyces capillispiralis* DSM 41695	31.5
BSE 7-9	*Streptomyces capillispiralis* DSM 41695	31.5
I5	*Streptomyces capillispiralis* DSM 41695	31.5
I4	*Streptomyces capillispiralis* DSM 41695	31.4
BSE 7F	*Streptomyces capillispiralis* DSM 41695	31.4

**Table 3 marinedrugs-19-00316-t003:** List of Indonesian actinomycetes strains with number and type of BGCs as predicted by antiSMASH analysis.

Strain	Total BGCs	PKS	NRPS	Hybrid BGC	Terpene	RiPP	Siderophore	Others
DHE 17-7	30	6	7	-	6	3	3	5
DHE 7-1	27	6	6	3	5	-	3	4
SHP 22-7	25	5	6	1	4	1	2	6
I4	19	3	2	-	4	4	2	5
I3	17	4	2	1	4	2	2	3
I5	19	3	2	1	4	4	2	3
BSE 7F	23	3	1	4	5	4	2	4
BSE 7-9	22	4	2	3	4	1	2	6
I6	24	3	6	1	2	1	2	9

**Table 4 marinedrugs-19-00316-t004:** Correlation between known compounds and BGC distribution in the nine Indonesian strains. A checkmark (√) indicates identified BGC in the studied strain, a question mark (?) indicates that BGC is not identified in the studied strain, and a minus sign (-) indicates the compound is not present in the medium.

Ion Cluster Name (Ion Formula)	*m/z* Measured	Adduct	Main Producer and Media Type		BGC Identified
			Solid	Liquid	
Ferrioxamine D1 (C_27_H_48_N_6_O_9_)	656.2830	[M − 2H + Fe]^+^	SHP 22-7; I3; I4; I6	-	√
Naphthyridinomycin A (C_21_H_28_N_3_O_6_)	418.1980	[M + H]^+^	I3; I4; I5	BSE 7F; BSE 7-9; I5	√
Amicetin (C_29_H_43_N_6_O_9_)	619.3100	[M + H]^+^	-	SHP 22-7; DHE 17-7	√
Antimycin A2 (C_27_H_39_N_2_O_9_)	535.2659	[M + H]^+^	-	BSE 7F	√
ECO-0501 (C_46_H_69_N_4_O_10_)	837.5022	[M + H]^+^	-	DHE 17-7	?
Echinoserine (C_51_H_69_N_12_O_14_S_2_)	1137.4504	[M + H]^+^	I6	-	√
Echinomycin (C_51_H_65_N_12_O_12_S_2_)	1101.4279	[M + H]^+^	I6	-	√
Tirandamycin A (C_18_H_25_O_6_)	337.1650	[M + H]^+^	I6	-	√
Staurosporine (C_28_H_27_N_4_O_3_)	467.2070	[M + H]^+^	I6		√

**Table 5 marinedrugs-19-00316-t005:** Overview of analogs and putative new compounds identified for the nine Indonesian *Streptomyces* strains. A minus sign (-) indicates that the compound is not present in the medium.

Ion Cluster Description	*m/z* Measured	Adduct	Main Producer and Media Type	
			Solid	Liquid
Ferrioxamine analogs	627.3303	[M − 2H + Fe]^+^	I3; I4; I6; DHE 17-7	-
	788.3753	[M − 2H + Fe]^+^	BSE 7-9	-
	840.4060	[M − 2H + Fe]^+^	-	BSE 7-9
	640.2520	[M − 2H + Fe]^+^	-	DHE 17-7
	654.2685	[M − 2H + Fe]^+^	DHE 17-7	-
Putative new peptides	598.2834	[M + 2H]^2+^	I3, I5, BSE 7F	-
	662.8048	[M + 2H]^2+^	I3, I5, BSE 7F	-
	727.3259	[M + 2H]^2+^	I3, I5, BSE 7F	-
Putative new compound group I	821.3349	[M + H]^+^	SHP 22-7	-
	734.3031	[M + H]^+^	SHP 22-7	-
	679.2430	[M + H]^+^	SHP 22-7	-
	647.2710	[M + H]^+^	SHP 22-7	-
Putative new compound group II	435.2774	[M + 2H]^2+^	-	DHE 17-7
	442.2857	[M + 2H]^2+^	-	DHE 17-7
	449.2934	[M + 2H]^2+^	-	DHE 17-7
	474.2833	[M + 2H]^2+^	-	DHE 17-7

## Data Availability

The complete genomes sequence data were deposited at the National Center for Biotechnology (NCBI) information data base, https://www.ncbi.nlm.nih.gov/genome (29 December 2020 for all, except SHP 22-7 (7 September 2018) and BSE 7F (4 May 2018)) with the accession numbers QEQV00000000 for BSE 7F, QXMM00000000 for SHP 22-7, SAMN15691494 for DHE 7-1, SAMN15691533 for I3, SAMN15691540 for I4, SAMN15691656 for I5, SAMN15691724 for BSE 7-9, SAMN15692265 for DHE 17-7, and RHDP00000000 for I6. GNPS job data: https://gnps.ucsd.edu/ProteoSAFe/status.jsp?task=429506a1cc2c4a679b421cc455c0249b (accessed on 12 March 2021).

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
