# Peer review of "Mining Indonesian Microbial Biodiversity for Novel Natural Compounds by a Combined Genome Mining and Molecular Networking Approach"

_marinedrugs, 2021, doi:10.3390/md19060316_

Round 1

Reviewer 1 Report

I greatly appreciated the enormous work that the authors have done writing the manuscript "Mining Indonesian microbial biodiversity for novel natural compounds by a combined genome mining and molecular networking approach". The authors attempted to described new strategies for reporting unknown compounds that may serve as a basis for further developments in this area of research.

Manuscript may be accepted with minor corrections

  1. 65-66: repetition of the previous sentence. Please rewrite/replace this statement.
  2. Table 2: could be more detailed in legends and heading
  3. Please improve the quality of figure 2.
  4. The incorporation of an experimental image of the zone of inhibition may improve the content.
  5. Kindly correct 16s to16s rRNA or rDNA.
  6. Kindly correct all the scientific names to italics throughout the manuscript.
  7. Results 2.6 section could be improved by adding some information about the consistency of the results (findings) with the recent findings of various similar works.
  8. About Figure 3, please describe the details about the distance between the selective species in Clade A-D.
  9. Please provide the scale bar for the Figure 1 panels. Scale bar 1 cm – How many kilometers?
  10. In figure 2, please elaborate on the incubation duration, 24 hr incubation? MIC values?
  11. Is the antibacterial effect reversible in either G+ or G-?

Overall, the manuscript is satisfactory and informative. I have given minor comments. Also, I would suggest thorough proofreading for rectifying grammatical and usage errors if any. I would recommend the publication of this manuscript after addressing minor changes.

Author Response

Comments to Reviewers

Reviewer 1

I greatly appreciated the enormous work that the authors have done writing the manuscript "Mining Indonesian microbial biodiversity for novel natural compounds by a combined genome mining and molecular networking approach". The authors attempted to described new strategies for reporting unknown compounds that may serve as a basis for further developments in this area of research.

Thanks a lot for the positive feedback on the manuscript! This is highly appreciated.

Minor issues are addressed below:

  1. 65-66: repetition of the previous sentence. Please rewrite/replace this statement.

That’s correct. Thanks for the hint. We have deleted this sentence

  1. Table 2: could be more detailed in legends and heading

Legend and heading of table 2 have been optimized

  1. Please improve the quality of figure 2.

Quality of Figure 2 has been optimized

  1. The incorporation of an experimental image of the zone of inhibition may improve the content.

Thanks for the suggestion. Unfortunately, we do not have a representative figure, which covers all performed bioassays. We saved all data in a separate Excel sheet, which we could provide but this is most likely not what the reviewer is asking for Thus, we did not include it so far but could do so if requested.

Kindly correct 16s to16s rRNA or rDNA.

Thanks for this attentive observation. This has been corrected throughout the manuscript

  1. Kindly correct all the scientific names to italics throughout the manuscript.

Thanks for the hint. We corrected this. Actually, this was not our fault – in our submitted version all the strain names were written in italics, however, the editorial office obviously sent a new template version to the reviewers without our consent which obviously contained these format errors. We have complained to the publisher accordingly

  1. Results 2.6 section could be improved by adding some information about the consistency of the results (findings) with the recent findings of various similar works.

We have added some additional information and discussion to results section 2.6 in the revised version

  1. About Figure 3, please describe the details about the distance between the selective species in Clade A-D.

Details on the polyphasic characteristics of the most closely related type strains of each clade are now given in the Suppl. Material part.

Clade A – Streptomyces luteus (according to Luo et al., 2017):

The strain was aerobic, Gram-stain-positive, with an optimum NaCl concentration for growth of 5 % (w/v). The isolate formed white aerial mycelium that was long filamentous with few branches; the substrate mycelium possessed long, smooth-surfaced spore chains bearing smooth spores and produced a yellow diffusible pigment. The strain contained iso-C16 : 0, anteiso-C15 : 0, anteiso-C17 : 0 and C16 : 0 as major cellular fatty acids. The predominant menaquinones of the strain were MK-9(H6), MK-9(H4) and MK-9(H10). The whole-cell sugar pattern contained glucose and ribose. The polar lipid pattern of the strain consisted of phosphatidylethanolamine, diphosphatidylglycerol, phosphatidylinositol, phosphatidylglycerol and phosphatidylinositolmannosides.

Clade B – Streptomyces capillispiralis (according to Mertz and Higgins et al., 1982):

Streptomyces capillispiralis produces a cephalosporin C-4 carboxymethyl esterase. The key characteristics of this species are gray spore mass color, spiral spore chains, and hairy spores.

Clade C – Streptomyces bungoensis (according to Eguchi et al., 2013):

Streptomyces bungoensis forms gray aerial mass, spiral spore chains, and a spiny spore surface; forms a melanoid pigment on tyrosine agar, on peptone-yeast extract-iron agar, and in tryptone-yeast extract broth; and has cell wall chemotype I.

Clade D – Streptomyces spongiicola (according to Huang et al., 2016):

The major menaquinones were MK-9 (H6) (65.6 %), MK-9 (H4) (23.8 %) and MK-9 (H8) (10.6 %). The predominant fatty acids were anteiso-C15 : 0 (25.5 %), iso-C16 : 0 (19.5 %) and iso-C15 : 0 (15.4 %). The predominant phospholipids were diphosphatidylglycerol, phosphatidylglycerol and phosphatidylethanolamine. In addition, four unidentified phospholipids were found. The G+C content of the genomic DNA was 69.8 mol%.

  1. Please provide the scale bar for the Figure 1 panels. Scale bar 1 cm – How many kilometers?

Scale bars for the Figure 1 panel have been added

  1. In figure 2, please elaborate on the incubation duration, 24 hr incubation? MIC values?

Additional details are added to the figure legend. Agar plugs were used after ten days of growth of the respective actinomycetes strains. Since these were bioassays data from agar plug assays MIC values cannot be given.

  1. Is the antibacterial effect reversible in either G+ or G-?

 We are very sorry but we are not sure if we have understood the question correctly. Does the reviewer mean if we observe antibacterial effect repeatedly against Gram+ and Gram- with the respective samples? Here the answer is yes. We have repeated the bioassays at least three times independently and obtained the same results. If the question is if samples, which show activity against Gram + are also active against Gram – and vice versa? Then the question is no; not always. For example, some samples are only active against Gram-positives but not against Gram-negatives.

Overall, the manuscript is satisfactory and informative. I have given minor comments. Also, I would suggest thorough proofreading for rectifying grammatical and usage errors if any. I would recommend the publication of this manuscript after addressing minor changes.

Thanks again for the valuable comments!

Indeed, we found that there were differences between our submitted version and the version that has been sent to the experts for revision. The Editorial Office has made changes in the format of the manuscript, which caused font errors and sent the final version to the reviewers without our consent. We apologize for this and have already complained to the editorial office. In the revised version we have corrected all errors carefully.

General comment

In addition to the reviewer comments, we have observed that the quality of Figure 5 was not optimal and have replaced it by an improved version.

Reviewer 2 Report

This work proposes a combination of genome mining analysis and mass spectrometry-based molecular networking to exploit Indonesia’s microbial diversity for actinomycetes towards the discovery of potential novel antibiotics. This study is of timely relevance. Although similar approaches have been presented before, there is still room for improvement and practical application. That being said, I have some improvement recommendations.

The introduction section is comprehensive in terms of the biological context and presents a clear, on-point rationale for the present proposal. However, a review of existing computational approaches is missing. Authors point to the Global Natural Product Social (GNPS) platform and claim that “… such platform iteratively proves its effectiveness to arrange seamlessly large numbers of samples enabling dereplication and tentative structural identification and/or classification…”. How exactly is effectiveness evaluated? No comparison is provided nor mention possible alternatives. For example, why not take advantage of SMIPS, using the predicted proteins as input? Likewise, works such as PMID: 28398567 would help readers to have a better bioinformatics perspective on the topic.

For the most part, the materials and methods and the results sections are well documented. Nevertheless, the authors do not link the parametrization of the different computational tools with the obtained results. In particular, it is relevant to discuss the obtained gene clusters in terms of how extensive/complete the in silico screening was and, eventually, how different parametrisation may render other potential results of interest. Likewise, figure captions should be as comprehensive and clear as possible. For example, in Figure 4, the authors state that colours denote BGC type, but since the patterns are labelled by name, what is the added value of the colour?

English quality is acceptable, but I recommend a thorough revision of the manuscript to solve inconsistencies and typos. For example, some italics are missing and values are sometimes written in numbers and others in texts.

Author Response

Comments to Reviewers

Reviewer 2

The introduction section is comprehensive in terms of the biological context and presents a clear, on-point rationale for the present proposal. However, a review of existing computational approaches is missing. Authors point to the Global Natural Product Social (GNPS) platform and claim that “… such platform iteratively proves its effectiveness to arrange seamlessly large numbers of samples enabling dereplication and tentative structural identification and/or classification…”. How exactly is effectiveness evaluated? No comparison is provided nor mention possible alternatives. For example, why not take advantage of SMIPS, using the predicted proteins as input? Likewise, works such as PMID: 28398567 would help readers to have a better bioinformatics perspective on the topic.

Thanks for this comment. Here we kindly ask the reviewer to note that this study does not intend to compare or review bioinformatic tools. We have applied the most comprehensive and well accepted tools in the field of taxonomy (TYGS), MS-networking (GNPS) as well as in the field of secondary metabolite genome mining (antiSMASH). Particularly, regarding the bioinformatics perspective, in our experience all given alternatives will even miss biosynthetic gene clusters. Thus, the presented genetic potential is at least according to todays standards completely evaluated.

For the most part, the materials and methods and the results sections are well documented. Nevertheless, the authors do not link the parametrization of the different computational tools with the obtained results. In particular, it is relevant to discuss the obtained gene clusters in terms of how extensive/complete the in silico screening was and, eventually, how different parametrisation may render other potential results of interest.

Thanks for this suggestion. However, we think that there are too less reliable data available to perform a comprehensive and reliable parametrization, so the obtained data would result in a rather hypothetical discussion. In order to partially meet the reviewer’s comment we have included some information and comparisons to similar molecular and genome mining networking studies, which is mentioned in paragraph 2.6 in the revised version.

Likewise, figure captions should be as comprehensive and clear as possible. For example, in Figure 4, the authors state that colours denote BGC type, but since the patterns are labelled by name, what is the added value of the colour?

Thanks for the hint. We adapted Figure 4 and legend accordingly.

English quality is acceptable, but I recommend a thorough revision of the manuscript to solve inconsistencies and typos. For example, some italics are missing and values are sometimes written in numbers and others in texts.

Indeed, we found that there were differences between our submitted version and the version that has been sent to the experts for revision. The Editorial Office has made changes in the format of the manuscript, which caused font errors and sent the final version to the reviewers without our consent. We apologize for this and have already complained to the editorial office. In the revised version we have corrected all errors carefully.

Many thanks for the positive assessment of the manuscript.

General comment

In addition to the reviewer comments, we have observed that the quality of Figure 5 was not optimal and have replaced it by an improved version.
